



# Development of a portable Cavity Enhanced Absorption Spectrometer for the measurement of ambient $N_2O_5$: experimental setup, lab characterizations, and field applications under polluted urban environment

**Haichao Wang[1], Jun Chen[2], Keding Lu[1]**

[1]State Key Joint Laboratory of Environmental Simulation and Pollution Control, College of Environmental Sciences and Engineering, Peking University, Beijing, China

[2]Institute of Particle and Two-Phase Flow Measurement, College of Energy and Power Engineering, University of Shanghai for Science and Technology, Shanghai, China

*Correspondence to:* K. Lu (k.lu@pku.edu.cn)

**Abstract.** A small and portable incoherent broadband cavity enhanced absorption spectrometer (IBBCEAS) for the dinitrogen pentoxide ($N_2O_5$) measurement has been developed. The instrument is featured with a mechanically aligned nonadjustable optical mounting system. To minimize the influence of the aerosol extinction and strong nonlinear absorption by water vapor, a dynamic reference spectrum

with NO titration is used for the spectrum analysis. The range of spectrum fitting is 640-680 nm. Moreover, the wall losses of $NO_3$ and $N_2O_5$ on the surface of the inlet and cavity tubes were extensively characterized. We determined that the surface reactivity for the heated PFA materials toward $NO_3$ and $N_2O_5$ is 0.16 s$^{-1}$ ± 0.04 s$^{-1}$ and 0.019s$^{-1}$ ± 0.004 s$^{-1}$, respectively. The $N_2O_5$ transmission efficiencies over the filter is 93% (± 3%). For the typical field experimental set up we used, the total transmission

efficiency is about 82.9%, the optimal limit of the detection (LOD) is estimated as 1.9 ppt (1σ) at 50 s intervals, the total uncertainty of the $N_2O_5$ measurement is 15%, which is dominated by the uncertainty of the $NO_3$ cross section calculated for 353K in this system. Our instrument has been successfully deployed in two comprehensive field campaigns conducted in northern rural areas of Beijing in 2016. In both campaigns, the new design of the optical mounting system enabled us a fast setup and stable running of

the IBBCEAS system for the detection of $N_2O_5$. High concentrations of $N_2O_5$ up to 1 ppb were detected for the two campaigns indicating an active nighttime chemistry presented in rural Beijing.

## 1    Introduction

Dinitrogen pentoxide ($N_2O_5$) is one of the most important reactive nitrogen species which plays an central role on the nighttime radical chemistry through its fast exchange with the $NO_3$ radical; meanwhile, the

heterogeneous uptake of $N_2O_5$ by ambient aerosols is one of the two most important removal pathways of NOx which affects the NOx budget from regional to global scale (Brown et al., 2012; Wang et al., 2015). In recent studies, uptake of $N_2O_5$ on the aerosols which contain chloride ions was found to produce considerable amount of $ClNO_2$ at night and show potential strong impact on the next day chemistry



through the production of Cl radical by the photolysis of ClNO$_2$ (Osthoff et al., 2008; Thornton et al., 2010). In the study of the N$_2$O$_5$ chemistry, the quantification and the detailed mechanism of the N$_2$O$_5$ uptake processes is the major gap to be filled in the future studies. However, the high NOx and high aerosol regions is the most suitable condition to carry on the study. As the satellite data showed, the US, Europe and China are the three major NOx regions (i.e. Richter et al., 2005) while the North China Plain

is the only one overlap with high aerosol loadings. Since extensively field studies of N$_2$O$_5$ were conducted in US and Europe (i.e. Brown et al., 2006; Crowley et al., 2010; Benton et al., 2010), direct observations of N$_2$O$_5$ for Chinese mega-city regions are of high value for the further exploration of the N$_2$O$_5$ chemistry.

There are a few existing optical spectroscopy and mass spectrometry methods for the detection of N$_2$O$_5$ such as cavity ring-down spectroscopy (CRDS), cavity enhance absorption spectroscopy (CEAS),

laser-induced fluorescence (LIF), and chemical ionization mass spectrometry (CIMS). With respect to CIMS, there are two possible methods utilizing different ionization reactions with different target ions like NO$_3^-$ (Slusher et al., 2004) and I(N$_2$O$_5$)$^-$ (Kercher et al., 2009). The method with NO$_3^-$ is susceptible to chemical interference like ClONO$_2$, BrONO$_2$ (Chang et al., 2011) and was found to have strong interference under high NOx regime in Hongkong (Wang et al., 2014). The method with I(N$_2$O$_5$)$^-$ is better

and showed good comparison versus a well-established CRDS system (Kercher et al., 2009; Wang et al., 2016). With respect to the optical approaches, a thermal decomposition plus a subsequent detection of the generated NO$_3$ radicals based on its absorption features in the visible spectral regions is required. (Yokelson et al., 1994). CRDS is the first technique used in the field measurement by detecting the N$_2$O$_5$ via its thermal conversion to NO$_3$ (Simpson et al., 2001), which has high temporal and spatial resolution

with high sensitivity and accuracy (Brown et al., 2002; Dube et al., 2006). In the comparison the temporal resolution and sensitivity of LIF is relatively low due to the low fluorescence quantum yield of NO$_3$ and is subject to interferences from NO$_2$ in the high NOx region (Matsumoto et al., 2005). According to intercomparison experiments under simulated conditions (Fuchs et al., 2012), the IBBCEAS technique (Fiedler et al., 2003; Ball et al., 2004; Langridge et al., 2006; Venables et al., 2006; Langridge et al., 2008;

Benton et al., 2010; Kennedy et al., 2011) showed similar detection capability of N$_2$O$_5$ to the CRDS system. Both CEAS and CRDS systems applied optical cavity, the CEAS measure the absorption spectrum of a wavelength window while the CRDS observed cavity decay time caused by the molecules at specific wavelength. Therefore, the CEAS shows a better selectivity while the CRDS shows a better sensitivity.

The measurement of N$_2$O$_5$ is still sparse in mainland China. The only published work was a field study performed at a mountain site in Hongkong (Wang et al., 2016; Brown et al. 2016), of which high N$_2$O$_5$ concentrations up to 8 ppbv were observed with two systems based on CIMS and CRDS, respectively. By considering the chemical complex environment presented in the Chinese City Clusters and the high NOx emissions, we think the selectivity may be more important. In addition, to probe the nighttime chemistry,

vertical profile measurement is more important than the study of the daytime chemistry since the ambient air is highly stratified at night (Stutz et al., 2004). Thus, a portable IBBCEAS system to detect ambient N$_2$O$_5$ concentrations was developed to carry out the study. This system is distinct from previous



$N_2O_5$-BBCEAS systems (Langridge et al., 2008; Kennedy et al.,2011) of its rigid cavity design and the usage of a dynamic zero point calibration system. This newly built instrument was deployed in two recent campaigns performed at two rural sites in Beijing. In this work, the detailed setup of our instrument, lab characterizations and its first field measurement results will be presented.

## 2 Instrument description

### 2.1 The Incoherent Broad Band Cavity Enhanced Absorption Spectroscopy

IBBCEAS, proposed by Fiedler et al. (2003), is a cavity-enhanced method which comprises a broad band light source, a high finesse optical cavity with high reflectivity mirrors, and a spectrograph with a CCD camera. The IBBCEAS technique in general combines the simplicity in the experimental setup, a high selectivity as the LP-DOAS method due broad spectra band and a high sensitivity with the long effective path length. This technique has been successfully utilized to measure a number of atmospheric trace gas compounds like HONO, $H_2O$, IO, $O_3$, $O_4$, $I_2$, IO, OIO, $SO_2$, $NO_3$, $N_2O_5$, glyoxal (CHOCHO) and methylgloxal ($CH_3COCHO$) (Washenfelder et al., 2008; Thalman and Volkamer, 2010; Gherman et al., 2008; Axson et al., 2011; Kahan et al., 2012; Washenfelder et al., 2013; Washenfelder et al., 2016; Min et al., 2016). The detection of $NO_3$ with this technique had been shown to be successful in the simulation chamber conditions with open-path IBBCEAS setup (Venables et al. 2006; Varma et al., 2009). Shortly afterwards, the closed cavity type of IBBCEAS was set up and shown to be successful in a surface measurement of $NO_3+N_2O_5$ (Langridge et al. 2008; Benton et al., 2010) and in a flight measurement of $NO_3$ and $N_2O_5$ (Kennedy et al. 2011).

The principle of IBBCEAS system in this work is similar as previous ones listed above and only introduced briefly here. The extinction coefficient, α, in the cavity can be described as Eq. 1 for the weak absorptions. On the one hand, α can be deduced by the measureable terms such as light transmission through the cavity, the mirror reflectivity and the effective cavity length; on the other hand, α is intrinsically caused by the absorption of the sample gases, the Rayleigh scattering of the sample gases and the Mie scattering of the aerosol from the sample gas.

$$\alpha(\lambda) = \left( \frac{I_0(\lambda)}{I(\lambda)} - 1 \right) \left( \frac{1-R(\lambda)}{d_{eff}} \right) = \sum_i n_i \sigma_i + \alpha_{mie} + \alpha_{rayl} \qquad (Eq.1)$$

Of Eq. 1, $\lambda$ is the wavelength of light, $n_i$ and $\sigma_i(\lambda)$ are the number density and absorption cross section of the $i_{th}$ gas compound which causes absorption of the incidental light, $d_{eff}$ is the effective cavity length, $R(\lambda)$ is the mirror reflectivity, $\alpha_{Ray}(\lambda)$ is the extinction due to Rayleigh scattering and $\alpha_{Mie}(\lambda)$ is the extinction due to Mie scattering, $I_0(\lambda)$ is the reference spectrum, and $I(\lambda)$ is the measured spectrum.

A least square spectral fitting method based on the IDL (Interface Define Language) is developed for retrieving molecule number densities of the species, the targets are the molecule number densities of $NO_3$ and $NO_2$, the molecule number densities of $NO_2$ is in sample of ambient air after thermal decomposed which is not exactly the same as ambient concentration. Spectra fitting window is from 640 to 680 nm, and a three-order polynomial was applied to fit the drift of light intensity and other scattering effect.



## 2.2 Optical layout

The instrument schematic layout is shown in Fig.1. The system comprises several parts: a temperature
stabilized light source, an optical cavity fixed on the aluminum profile and a commercial spectrograph
with CCD detector. A single light emitting diode (LZ1-00R200, LedEngin, Marblehead, MA, USA) was
used as a light source. It provided 1000 mW nominal optical power centered at deep red light region (662
nm) and the full width at half maximum (FWHM) is 25 nm when the LED temperature is stabilized at
17.5±0.1 °C to minimize the emission spectral shift by using a heat sink with thermo-electric cooler (TEC)
control module . The LED is also mounted on a 3 dimension (3D) adjustable bracket which can be
adjusted to improve the cavity alignment.

The optical cavity designed to be pressure and temperature insensitive. Therefore high reflectivity cavity
mirrors are mechanically aligned with high precision mirror mounting parts based on pilot experiments.
In the pilot experiments, the distances and positions of different mirrors/lens are tested and optimized to
achieve maximum light output of the cavity. The advantage of this mirror mounting design is the high
stability and fast setup of the cavity system for the purpose of the field measurements. The disadvantage
is that one mirror mounting system is only suited best for specific type (wavelength region, thickness and
curvature et al.) of cavity mirrors. In the current setup, the optical cavity consists of a pair of high
reflective (HR) mirrors. The peak reflectivity of the mirrors at 660 nm is of about 0.999936 with the radii
of curvature of $100 \pm 5$ cm (Layertec GmbH, Mellingen, Germany), the diameter of the mirrors is 25.0
(−0.1) mm. The two mirrors is mounted on two opposite located customized lens tubes, which are
separated by 50.0 cm. Each HR mirror is continuously purged by 100 ml/min pure nitrogen flow to
prevent the particle pollution enrolled by the sample gas flow. The optical cavity is enclosed by a sample
gas flow tube and two corrugated pipes. The ambient air is sampled through an aerosol filter into the flow
tube so that the aerosol extinction can be reduced significantly. This design is especially important for the
use of such system for high aerosol loading environment. The corrugated pipes are used to buffer the
length change of the sample flow tube under heated conditions. The light source and the optical
components are homocentric integrated on an aluminum profile ($75 \times 8 \times 5$ cm). A 30 mm focal length
achromatic lens is installed the lens tube 1 (L1) near the light source side as collimating lens. On the other
side of the cavity, a 50 mm focal length achromatic lens is installed in the lens tube 2(L2) after the cavity
HR mirror which further coupling the transmitted light onto a 100 μm diameter, 0.22 numerical aperture
optical fiber. The fiber is mounted on a 3D adjustable bracket to improve the fiber coupling through
relative movement toward the 50 mm lens. Finally, the fiber directed the cavity output light into the
Ocean Optics QE65000 spectrometer (Dunedin, FL, USA). The charged couple device (CCD) in the
QE65000 spectrograph is thermally regulated at −20.0 °C to minimize dark current. The line density of
diffraction grating of the spectrometer is 1200 mm$^{-1}$, the entrance slits width is 100 μm and the spectral
resolution FWHM is 0.85 nm with the wavelength coverage 580-710 nm. The instrument is working
under the SNR (Signal Noise Ratio) estimation larger than 500:1.

## 2.2 Flow system

The instrument sample flow system included the inlet tube, aerosol filter, thermal dissociation module
and heated sample flow tube, purge flow, mirror reflectivity calibration module, NO titration module and



air mass sensors (T, P and RH).

The sample gas flow tube is made by a 35.6 cm long perfluoroalkoxy polymerresin (PFA) coated stand steel tube, the inner diameter (ID) is 10.0 mm. One end of the cavity connected a PFA inlet interface and the other end connected the outlet interface, the inlet/outlet interface is 1.8 cm long and the ID is also 10.0 mm, the distance between the inlet and outlet is 37.4 cm, but the total length of the PFA inner coated is 39.2 cm. With this combination, the loss of $NO_3$ during detection is minimized.

A Teflon polytetrafluoroethylene (PTFE) membrane filter (25 µm thickness, 4.6 cm diameter, 2.5 µm pore size) is used to remove ambient aerosol and protect the HR mirrors. A 35 cm long 1/4 inch ID PFA tube is installed between the filter and the inlet interface. This preheater tube is heated and stabilized at 120 °C serving as a thermal dissociation reactor for $N_2O_5$. When the sample flow rate is 2.0 L/min, the residence time in the thermal dissociation tube is about 0.13 s. With this setup of temperature and residence time, $N_2O_5$ will be completely decomposed to $NO_3$ in front of the preheater tube. The sample gas flow tube is heated and stabilized to 80 °C to prohibit the reversed reaction of $NO_3$ and $NO_2$ into $N_2O_5$. Between the filter and the thermal dissociation chamber, the NO titration module is inserted by a tee-piece PFA to add the NO gas. Details of the NO titration module will be given in Sect. 3.4. The mirror reflectivity calibration is done by calculation the difference of the Rayleigh extinction coefficient between pure $N_2$ and He gases (Washenfelder et al., 2016; Min et al., 2016). For our system, the calibration gases are introduced into the cavity through the mirror purge flow lines.

## 3   Characterization of the instrument key parameters

According to the Eq.1, there are several key parameters can affect the precision and accuracy of the measurement including: the cross section of the target species, effective cavity length, mirror reflectivity and reference spectrum.

### 3.1 The absorption cross section ($\sigma_i(\lambda)$)

To retrieve molecule number densities from the Eq. (1), the effective absorption cross section of the abundant ambient absorbers is necessary in this wavelength window such as $NO_3$, $NO_2$ and $H_2O$. However, the $H_2O$ molecule vibration and rotation line-strength for a moderate spectrograph to resolve and in variable humidity conditions is still a challenge task, in this work to bypass this problem by using a reference spectrum which contains the same amount of $H_2O$ as the measurement spectrum. This is achieved by using NO titration for the setup of the reference spectrum (details see Sect. 3.4). Therefore this method enables a fast data processing during a field campaign which could be benefit for providing online measurement readings.

The $NO_3$ absorption cross section is known as temperature dependent (Wangberg et al., 1997; Sander, 1986; Ravishankara and Mauldin, 1986; Yokelson et al., 1994; Orphal et al., 2003; Osthoff et al., 2007). The $NO_3$ absorption cross section at standard condition is referred to Yokelson et al. (1994). For the heated cavity conditions (353K), the cross section of the $NO_3$ were calculated by two steps: first, scaling the $NO_3$ cross section profile from Yokelson et al. (1994) to the band's peak intensity at 662 nm at 353 K from Osthoff et al. (2007); secondly, determining the instrument absorption cross section by convolving



the scaled cross section with an instrument function (taken from the neon emission line at 659.48nm).
Fig.2 is the result of the scaled and convolved $NO_3$ absorption cross section at 353 K and the convolved
$NO_2$ absorption cross section. The absorption cross section of $NO_3$ near the 662 nm at 353K is three
orders of magnitude larger than that of $NO_2$ while the ambient $N_2O_5$ concentration is normally two orders
of magnitude smaller than that of $NO_2$ during nighttime, so the cavity detected $NO_3$ absorption near 662
nm will dominate the total absorption at night.

### 3.2 The effective cavity length ($d_{eff}$)


The experimental determination of the total extinction coefficient requires the knowledge on the effective
cavity length $d_{eff}$, which represented the cavity length occupied by the absorbing gas sample when the
sampling flow is stable. Since the continuously purge flow occupied the two ends of the cavity to protect
the mirrors, the $d_{eff}$ usually shorter than the distance between the two high reflective mirrors and longer
than the distance between the sampling inlet and outlet. We determined the $d_{eff}$ to be 45.0 cm for our
cavity setup with standard gas of $NO_2$. The determined $d_{eff}$ is larger than the length of the PFA coated part
with 39.2 cm. Since the wall reactivity of $NO_3$ toward none PFA coated material could be quite large, we
estimated the $d_{eff}$ for $NO_3$ detection in our system to be 39.2 cm. different $d_{eff}$ numbers were determined
for $NO_2$ and $NO_3$, our solution is to apply the $d_{eff}$ of $NO_3$ in Eg. (1) by adding a correction factor on the
used $NO_2$ absorption cross section.

### 3.3 The mirror reflectivity ($R(\lambda)$)

The mirror reflectivity ($R(\lambda)$) is an important parameter to be determined in this instrument. In previous
work, $R(\lambda)$ had been determined through the detection of stable trace gas compound with known
concentrations (Venables et al., 2006), the differentiation of pure gases with distinct Rayleigh Scattering
cross section (Chen and Venables, 2011; Washenfelder et al., 2016; Min et al., 2016), the use of low loss
optics (Varma et al., 2009) and phase shift method (Langridge et al., 2008; Kennedy et al. 2011). In this
study, measuring the pure $N_2$ and He Rayleigh scattering signals in the cavity through Eq. (2) is used to
determine $R(\lambda)$ during field campaigns. The Rayleigh scattering cross sections for $N_2$ ($\alpha_{Ray,N2}(\lambda)$) and He
($\alpha_{Ray,He}(\lambda)$) are referred to Sneep and Ubachs (2005) and Shardanand and Rao (1977), respectively.

$$R(\lambda) = 1 - d \left( \frac{I_{N_2}(\lambda)n_{N_2}\sigma_{Rayl,N_2}(\lambda) - I_{He}(\lambda)n_{He}\sigma_{Rayl,He}(\lambda)}{I_{He}(\lambda) - I_{N_2}(\lambda)} \right)$$    (Eq. 2)

Of Eq. (2), $d$ is the distance between the two high reflective mirrors, $I_{N2}(\lambda)$ and $I_{He}(\lambda)$ represent the $N_2$
and He spectrum respectively, which were acquired by stopping the sampling flow and filling the $N_2$ or
He with the cavity though the purge flow, $n_{N2}$ and $n_{He}$ are the number density of $N_2$ and He calculated by
the measured the temperature and pressure in the cavity.

The Fig.3 shows the mirror reflectivity calculation result from the University of Chinese Academy of
Science (UCAS) international field winter campaign, 2016. The bold black line is the averaged
reflectivity of the five measurement of $R(\lambda)$, noted that the peak of the $R(\lambda)$ is 0.999936 ± 0.000002 at 662
nm. Under the protection of the purge flow and due to the rigid setup of the cavity system, the $R(\lambda)$ was





remarkably stable during the field campaign. The bold red line is the averaged cavity loss, which is equal

to the $(1-R (\lambda))/d$, the maximized point near 662 nm is $(1.28 \pm 0.01) \times 10^{-6}$ (1σ). The effective path length calculated at 662 nm is 6.13 km.

### 3.4 The reference spectrum ($I_0(\lambda)$)

In this work, two kinds of reference spectrum $I_0(\lambda)$ are recorded for the analysis of the ambient dataset. One reference spectrum is obtained by measuring pure $N_2$ gases; the other is obtained by adding NO into

the ambient sample flow. When the reference spectrum with pure $N_2$ is applied to Eq. (1), we can retrieve $NO_2$, $NO_3$, and $H_2O$ concentrations in the targeted wavelength window. Nevertheless, the absorption of the water vapor near 662 nm is a typical non-Beer–Lambert absorption, its absorption lines in the 4v+δ polyad have pressure broadened half widths, which is much smaller than the resolution of spectrograph used in this work, so the effective cross section need to calculated frequently. Although there are several

ways to fitting the water vapor absorption very well in the measurement as demonstrated in previous publications (Ravi et al., 2009; Langridge et al., 2008; Kennedy et al. 2011), fitting the water vapor absorption is not the purpose of this instrument. When the reference spectrum with NO addition is applied to Eq. (1), the calculated extinction spectrum would be quite clean of which only contain the positive absorption of $NO_3$ and negative absorption of $NO_2$. The same concept has been used by the

CRDs for the detection of $NO_3$ and $N_2O_5$ (Brown et al., 2002). In current setup, the NO addition is modulated by a computer controlled solenoid valve, the flow of 98 ppm NO mixture in the $N_2$ is added with 10ml/min by a Teflon tube (O.D. = 1/4 inch), the excess NO is enough to chemical destructed the decomposing produced $NO_3$ completely during the 0.13 s residence time in the preheater tube, a dry $N_2$ line is added at the exit end of the solenoid valve by a PFA tee-piece to promote the diffusion of the

residual NO at the beginning of the NO off-mode, as shown in the Fig. 4, the 1/8 inch OD Teflon tube of the $N_2$ flow inserted into the 1/4 inch OD Teflon tube in a contrary direction with a continuous small flow rate (10ml/min), in a typical measurement cycle of 5 min, NO is added into the sample flow for 20 s with 5s spectra integral time so that a kind of dynamic zero point can be used for the $N_2O_5$ detection. An example of the time series of ambient $N_2O_5$ detection for half an hour is shown as Fig. 5. The red points

mark the effective ambient measurement result, which covered 4 minutes 20 s in every 5 minutes, the blue points include the 20 s background points and the 20 s ambient measurement points affected by the residual NO in the tube.

## 4    Result and discussion

### 4.1 Spectrum fitting

Fig. 6 shows an example of $NO_3$ spectrum fitting of an ambient measurement result at 5s. By using the reference spectrum given by the NO titration scheme described in Sect. 3.4, the fitting of $NO_3$ spectra is therefore a straight forward procedure. For most of the cases, $NO_3$ is the only target species during spectra process. Within the fitting window 640–680 nm, the fitted line in the Fig. 6a is corresponding to 74 ppt of $N_2O_5$ and the fitted line in the Fig. 6b represents 25ppt $N_2O_5$ in the cavity, the residuals of

fitting for both cases are shown as the Fig. 6c. The fitting spectrum residual is in the range of $\pm 4.0 \times 10^{-9}$



$cm^{-1}$ and absorption structure of $NO_2$ and $H_2O$ is not observed in the residual spectrum.

## 4.2 Transmission efficiency of $N_2O_5$

For the accurate measurement of $N_2O_5$, the $N_2O_5$ wall loss in the sampling manifold and the $NO_3$ wall loss in the preheater and cavity need to be determined. To quantify those two loss processes, we defined two transmission factors of $N_2O_5$ correspondingly. Transmission factor 1 (T1) characterize the survived part of $N_2O_5$ from the inlet to the beginning of the preheater, which caused by the wall loss on the sampling tube and the filter loss on the filter holder, transmission factor 2 (T2) characterize the survived part of the produced $NO_3$ from $N_2O_5$ from the beginning of the preheater to the center of the cavity outlet. To determine these two factors, a $N_2O_5$ source is set up through a 160 L flow reactor. The known amount of gases of $NO_2$ and $O_3$ are generated by a gas calibrator (Thermo146i) and then delivered into flow reactor. Stable steady state concentration (±2%) of $N_2O_5$ can be built up after passing this reactor. Through modulate the ratio of supplied $NO_2$ and $O_3$; the ratio of $N_2O_5$ to $NO_3$ can be typically controlled to more than 100:1.

### 4.2.1 Wall loss of $N_2O_5$ in the sampling manifold

Two losses processes contributed to the determination of T1, one is the loss of $N_2O_5$ on the particle filter and the other is that on the inlet tubes. Following Brown et al. (2002), we attached a three-way Teflon valve at the inlet and the standard $N_2O_5$ gas flow is switched between two sample lines with and without a clean particle filter. Differentiation of the above two $N_2O_5$ measurement results determines the transmission efficiency through particle filter which is 93.0% for our typical instrument parameters, the frequent filter changing keeps the correction factors with small uncertainty (± 3%), which is similar to previous results (Brown et al., 2002; Fuchs et al., 2008).

To test the wall loss of the $N_2O_5$ in the PFA line, different lengths (0.5, 3.5, 5.5, 7.5, 10.5 m) of PFA tube (O.D. =1/4 inch) is inserted between the outlet of the reactor and the inlet of the preheater. The apparent first order loss rate of $N_2O_5$ (0.015 $s^{-1}$) is deduced by a linear fit of the observed $N_2O_5$ concentrations toward different the residence times corresponding to certain length of tubes (Fig. 6). The actual situation is more complicated in these sample lines that contains a series of reactions (R1-R5). And therefore the wall loss rate of $N_2O_5$ is further abstracted from the observed decay rate with a box model including R1-R5. In this model, initial $NO_2$ and $O_3$ are observed values at the outlet of the reactor; wall loss rate of $NO_3$ (R3) is set to be 0.16 $s^{-1}$, which was determined by another experiment explained later. The retrieved net $N_2O_5$ loss rate coefficient was 0.019 $s^{-1}$ ± 0.004 $s^{-1}$, this result is smaller than the measured upper limit result 0.042 $s^{-1}$ ± 0.003 $s^{-1}$ by Kennedy et al. (2011). The typical 1.5 meter length of inlet sampling tube have the $N_2O_5$ transmission efficient at 98.9%. Transmission factor 1, T1, is therefore calculated to be 90.0% (±3.0 %) for the typical condition.

$N_2O_5 \rightarrow NO_3$ (R1)

$N_2O_5 + wall \rightarrow$ (R2)

$NO_3 + wall \rightarrow$ (R3)



$$NO_2 + O_3 \rightarrow NO_3 \qquad (R4)$$

$$NO_2 + NO_3 \rightarrow N_2O_5 \qquad (R5)$$

### 4.2.2 Wall loss of $NO_3$ produced from $N_2O_5$ in the heated tubes

When the $N_2O_5$ goes into the preheater, wall loss of $NO_3$ produced from $N_2O_5$ thermal decomposition become the dominant contributor, which limited by the transmission factor 2. To determine the wall loss reactivity of $NO_3$, the optical cavity is used as a flow tube. Stable amount of $N_2O_5$ is built up in the optical cavity with a continuous flow mode. When stopping the flow, the observed $NO_3$ versus the elapsed time determines the first order loss rate of $NO_3$ ($0.22$ s$^{-1}$ $\pm$ $0.04$ s$^{-1}$) in the optical cavity tube (see

Fig. 8). The fitted first order uptake coefficient of $NO_3$ reflects the contribution from three processes: (1) the wall loss of the $NO_3$ in the cavity; (2) the dilution effect due to the purge flow in the cavity; (3) the production of $NO_3$ with available $NO_2$ and $O_3$. The third term is minimized by setting the $NO_2$ and $O_3$ concentrations to be small values (in our case, we used 35 ppb for both of them). The second term can be calculated according to the sample flow and purge flow rates, and it can be independently determined

with the detection of $NO_2$ decay in the optical cavity after stopping the flow. With both methods, the dilution reactivity of our experimental setup determined to be $0.087 \pm 0.02$s$^{-1}$. The wall loss reactivity of $NO_3$ is retrieved to be $0.16 \pm 0.04$ s$^{-1}$ with a box model taking into account of the above three processes, which is in the range of the previous results with $0.1$-$0.3$ s$^{-1}$ (Brown et al.,2002; Crowley et al. 2010; Kennedy et al. 2011; Wang et al., 2015). Since the residence time in the preheater and cavity is 0.13 s and

0.83 s, the $T_2$ was calculated to be 92.1%. The total transmission efficiency of $N_2O_5$ ($T_{N2O5}$) can be calculated by $T_1 \times T_2$, was found to be 82.9%. Additionally, the total transmission efficiency of the ambient $NO_3$ ($T_{NO3}$) is calculated to be 63.6% when ignoring the $NO_3$ loss on the filter (Fuchs et al., 2008). These factors are suitable to be applied for the sampling by changing filter frequently.

### 4.3 Precision and uncertainty analysis

Fig. 9 shows the histogram of 5200 zero measurement results in the laboratory with 1 second time resolution, there exist an offset with -0.6 ppt and the limit of detection of 3.0 ppt (1$\sigma$). The limit of the detection and the stability of the instrument are further analyzed with Allan Variance method (Allan, 1966; Werle et al., 1993). Fig. 10 shows an Allan Variance analysis of the 12000 zero measurement spectrums in the laboratory with 1s integration time. According to the Allan deviation plot, the best limit of the

detection is found to be 1.9 ppt (1$\sigma$) at 50 s intervals. The uncertainty of the instrument consists of the following parts: the uncertainty of the temperature corrected and convolved cross section of $NO_3$ is $\pm$ 13% from the same method of Kennedy et al. (2011); the effective cavity length calculation is $\pm$ 2%; the reflectivity determination uncertainty mainly controlled by the cross section of $N_2$ and He, which is about $\pm$ 5% together; the uncertainty of the transmission efficiency of the sampling and cavity system is about $\pm$

3%, according to the Gaussian error propagation, the total uncertainty of $N_2O_5$ detection is estimated to be $\pm$ 15% in our system.



## 5 Instrument Performance in two comprehensive field campaigns

After the development and full characterization performed in the lab, our instrument had been successfully applied in two comprehensive field campaigns in 2016. The first campaign took place at the
campus of University of Chinese Academy of Sciences (UCAS) from 6 January to 4 March while the second campaign took place at Peking University Changping campus from 15 May to 23 June 2016. As shown in Fig. 11, both sites are located in northern rural areas in Beijing, of about 60 km and 40 km to the center of Beijing city, respectively. According to our current understanding of the $NO_3$-$N_2O_5$ chemistry, rural areas are the transitional regions of the anthropogenic and biogenic emissions where high
$O_3$ can meet with relatively high $NO_2$ so that the $NO_3$-$N_2O_5$ chemistry may be maximized. We expected these two sites to be ideal locations to probe the $NO_3$-$N_2O_5$ chemistry in Beijing.

During the UCAS campaign, our instrument was deployed at a roof lab and the sampling inlet was about 15 m above the ground. The measurement site is close to mountainous area in Beijing while influenced by nearby traffic emissions. When the northerly wind appeared, we sampled clean air masses
entrained with local traffic and residential emissions; when southerly wind appeared, we could then capture the outflow of Beijing. In Figure 12(a), the observed $N_2O_5$ concentrations during a typical development of such air mass change from clean to polluted conditions are shown. It is worth to be noticed that the observed $N_2O_5$ also including the ambient signals from $NO_3$ as described in Eq. 3. Of Eq. 3, $T_{NO3}$ is much smaller than $T_{N2O5}$ as discussed in Sect. 4.2.2. Moreover, due to the conditions we
experienced for the winter campaign, the ratio of $N_2O_5/NO_3$ is always larger than 10 so that this term $T_{NO3} \times C_{ambi}(NO_3)$ is ignored. For this reason, our detected $N_2O_5$ concentrations will have a positive bias less than 10%.

$$C_{obs}(N_2O_5) = T_{NO3} \times C_{ambi}(NO_3) + T_{N2O5} \times C_{ambi}(N_2O_5) \qquad \text{(Eq.3)}$$

High concentrations of $N_2O_5$ were observed at a near surface level at the UCAS site. As the development
of the pollution episode, the maximum $N_2O_5$ concentrations even reached to be more than 1ppb in the night of Mar 02-03, 2016. Fast variation of $N_2O_5$ was also observed which might due to local traffic emissions during stagnant conditions. In all these days, the observed $N_2O_5$ continuous to accumulate in a few hours after sunset, reached its maximum before midnight and then gradually decreased to zero before sunrise. The decrease of the $N_2O_5$ concentrations at night in this location may be related to the typical
running style of the heavy-duty vehicles (HDV). Typically, more heavy-duty cars were appeared on the nearby street after 22:00 since the ban of HDV entering into downtown Beijing is set to be after midnight. In addition, fast variation of the ambient $N_2O_5$ concentrations was captured indicating that the $N_2O_5$ distribution in the ambient air masses observed at UCAS was quite heterogeneous. This behavior has also been reported by many other field measurements (Brown et al., 2003; Slusher et al., 2004; Matsumoto et
al., 2005; Ayers et al., 2006; Nikayama et al., 2007).

During the PKU-CP spring campaign, our instrument was set up on the fifth floor of the main building in PKU-CP campus. Our inlet was about 15 m above ground. During springtime in northern part of rural Beijing, high $O_3$ event would be presented (e.g. Wang et al., 2006). Together with the atmospheric



processes with high $NO_2$ conditions, high $NO_3$ and $N_2O_5$ concentrations can be expected. At this location,
a steady state calculation showed that the dominant ratio of $N_2O_5/NO_3$ was again larger than 10 so that
our direction observations again showed that there were high $N_2O_5$ concentrations presented at evening
hours (Figure 12(b)). During the campaign, an Aerodyne $I^-$CIMS (Breton et al., 2012;Breton et al., 2014)
was deployed in parallel with our instrument. A preliminary comparison showed that there were good
agreements of the observed $N_2O_5$ between the two instruments. Details of this comparison will be
presented in a future publication.

## 6     Conclusions

Detection of $N_2O_5$ is an attractive and challenge work to explore the atmospheric oxidation capacity and
the NOx removal at night. $N_2O_5$ chemistry is very active and important for areas with the presence of
high $O_3$ and high $NO_2$ as well as high aerosol loadings. The Chinese megacities are these kind of areas
which fulfill such requirement and are expected to be the global hot spots of $N_2O_5$ chemistry. The cavity
based absorption techniques (e.g. CRDS and CEAS) and CIMS are the available technical options for the
field detection of $N_2O_5$.

In the present work, we developed a new portable $N_2O_5$ spectrometer featured with a mechanically
aligned lens tubes for the setting up of HR mirrors. The new design offers us a fast setup of the
instrument in the field and the instrument are proved to be running stably for a few months in two field
campaigns. Except the new design of the HR mirror mounting parts, a few important engineering work is
explored during setting up of the instrument. Temperature control of the light source (e.g. LED in our
case) is of crucial importance for the subsequent spectrum analysis. The using of the corrugated pipes in
between the mirror-mounting lens tubes and the cavity tube is critical to make the system to be stable for
the heated conditions. Since the mirror-mounting lens tubes are mechanically fixed, three-dimensional
adjustment of the light source and the fiber receiver is required to maximize the optical signal collected
by the detector.

In the lab, this work systematically characterized the key parameters of our $N_2O_5$ spectrometer such as
the mirror reflectivity and effective cavity length. The mirror reflectivity is found to be larger than 99.99%
as stated by the producer and the determined results were variable within 2%, which again showed the
advantage of this mechanically, aligned HR mirror-mounting lens tubes. A dynamic reference spectrum
generated by adding NO into the sample flow is tested and proved to be very helpful for the ambient
spectrum analysis that avoided the complicated fitting of $H_2O$ absorption. The wall losses of the inlet
system and the cavity tubes were also extensively characterized. The wall loss rate of the $N_2O_5$ in the
sampling tube is determined to 0.019 $s^{-1}$ ±0.004 $s^{-1}$ and the wall loss rate of the $NO_3$ radical produced
from thermally decomposed $N_2O_5$ in the preheater tube and the cavity tube is 0.16 $s^{-1}$ ±0.04 $s^{-1}$, the
transmission efficiency of $N_2O_5$ on the filter is 93(± 3)%, based on the mentioned parameters above, the
total $N_2O_5$ loss on the surface of the filter and the total transmission efficiency is about 82.9%, the
calibration factor used to calibrate the measured concentration to the ambient concentration is 1.20, and
the total accuracy of the measurement is determined to be ± 15%. The best fitting wavelength window is



found to be 640–680 nm, and the measurement precision is quantified to be 3.0 ppt ($1\sigma$) in 1 second and the Allan deviation analysis indicated that the best detection limit could be achieved as 1.9 ppt ($1\sigma$) at 50 s intervals.

    Our $N_2O_5$ spectrometer had been successfully applied in the field measurements of $N_2O_5$ during the

2016 UCAS winter campaign (Jan - Mar) and the 2016 PKU-CP spring campaign (May - Jun). In the example time series we presented herein, high and highly variable concentrations of near surface $N_2O_5$ concentrations were detected for Beijing rural areas. These high near surface $N_2O_5$ concentrations indicating that there were very active nighttime $NO_3$-$N_2O_5$ chemistry as well as the daytime OH chemistry (Lu et al., 2013; Lu et al., 2014; Tan et al., 2016) in the North China Plain. In a recent source

term diagnosis of the secondary organic aerosols, it is found that the nighttime $NO_3$ oxidation could even dominate the formation of organic nitrates over Europe (Kiendler-Scharr et al., 2016). Since all the Chinese megacity areas are the NOx hot spot worldwide, it will be therefore of crucial importance to bridge the daytime OH chemistry and the nighttime $NO_3$ chemistry in the near future in China. Our current developed portable $N_2O_5$ instrument and a planned portable $NO_3$ instrument can serve as the

useful tool to explore the formation of the atmospheric oxidation capacity in such areas.





*Acknowledgements.*

The work was supported by the National Natural Science Foundation of China (Grant No. 41375124,
21522701, 21190052, 91544225), Strategic Priority Research Program of the Chinese Academy of

Sciences (grant no. XDB05010500). The authors gratefully acknowledge the discussions and suggestions
from Steven Brown, Kyung-Eun Min, Bin Ouyang, Ravi Varma, Hendrik Fuchs and Zhiguo Wu. We
thank the team of the UCAS (organized by Yuanhang Zhang) and Changping Campaigns (organized by
Min Hu and Mattias Hallquist).

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

K and the red thick line is the scaled and convolved cross section at 353 K.





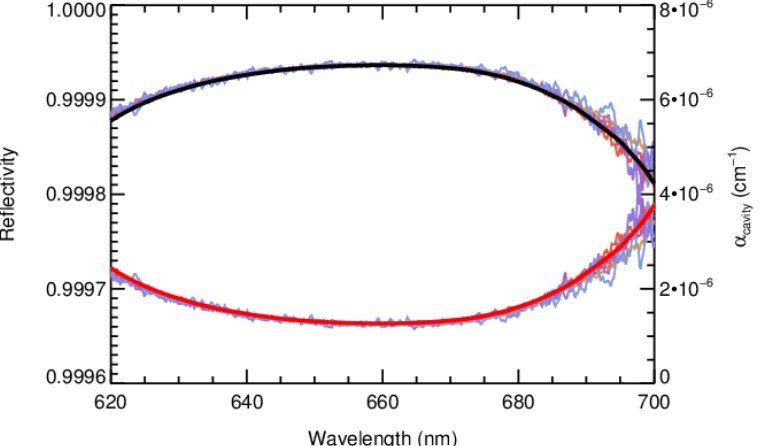


**Figure.3** Mirror reflectivity and the cavity loss calibrated with He/N2 measurement in current experimental setup during the UCAS campaign.   The original He/$N_2$ measurements during the UCAS campaign were depicted by varying colored lines, the smoothed black bold line is the averaged $R (\lambda)$ and the smoothed bold red line is averaged cavity loss $(1 - R (\lambda))/d$ ) from five measurements.  The mean ($\pm 1\sigma$) value at 662 nm of reflectivity and the cavity loss is $0.999936 \pm 0.000002$ and $(3.534 \pm 0.016) \times 10^{-6}$, respectively, the effective path length

at 662 nm reached 6.13 km.

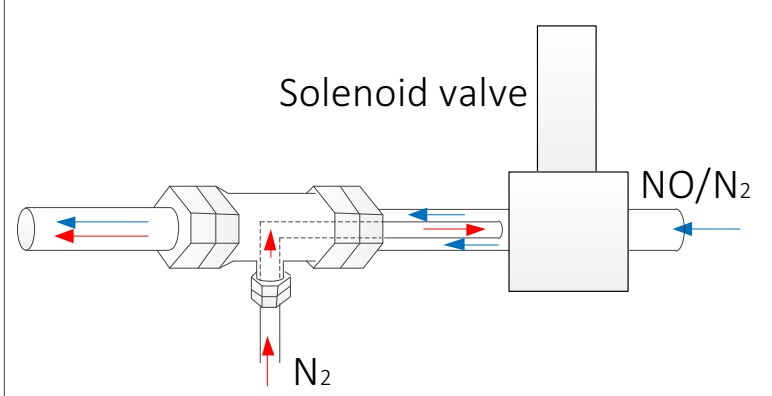

**Figure.4** The schematic layout of the NO addition module, the red arrow denote the $N_2$ gas flow and the blue arrow denote NO gas flow.





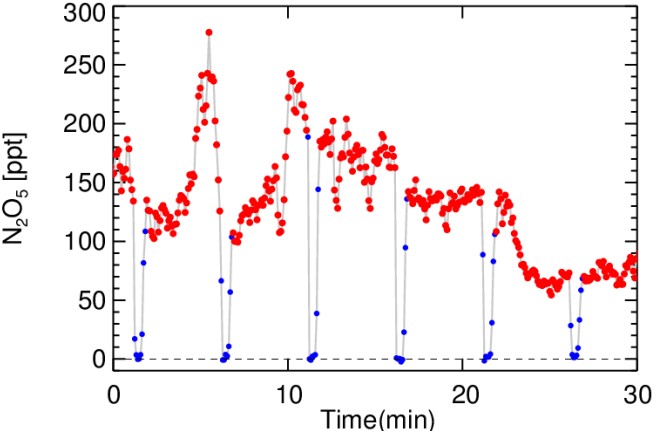

**Figure.5** Example time series of ambient $N_2O_5$ detection performed at a rural site in Beijing in 5s spectrum integral time. The red points

denote the ambient measurements without addition, the blue points denote the 20 s reference measurements without addition NO and 20 s

transition period measurement without addition NO.

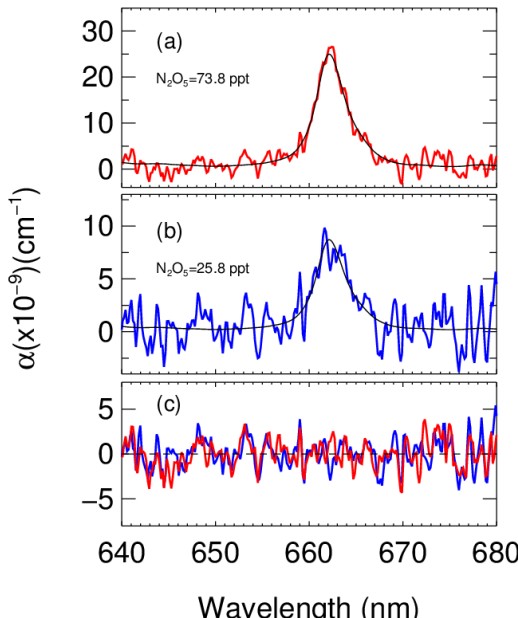

**Figure.6** Example of spectral fit of $NO_3$ radical in the heated cavity of the instrument for two ambient extinction spectrums measured in the

ambient environment.



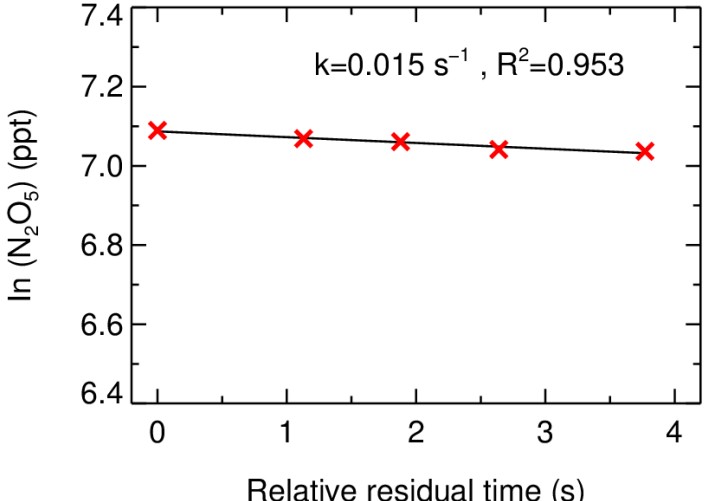

**Figure. 7** The decay rate of $N_2O_5$ in the PFA tubes. Red points denote the observation results and the black line depicts the corresponding fit.

The net wall loss reactivity of $N_2O_5$ is retrieved to be $0.019 \pm 0.004$ $s^{-1}$ with a box model taking into account of the chemical equilibrium and

purge flow dilution effect.

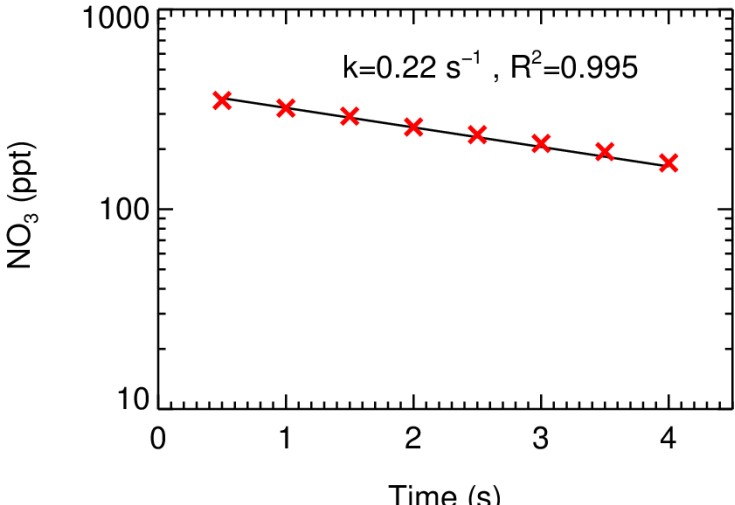

**Figure. 8** The decay rate of the $NO_3$ radical in the heated optical cavity. Red points denote the observation results and the black line depicts

the corresponding fit. The net wall loss reactivity of $NO_3$ is retrieved to be $0.16 \pm 0.04$ s-1 with a box model taking into account of the

chemical equilibrium and purge flow dilution effect.



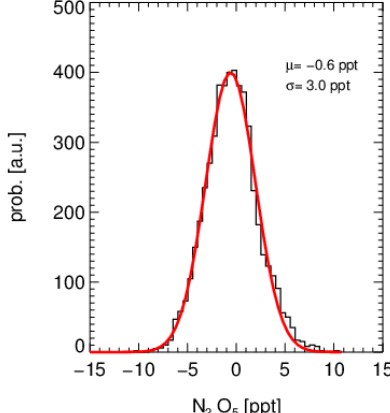

**Figure. 9** The histogram of 5200 zero measurement results in the laboratory, the mean values is -0.6 ppt with the limit of detection of 3.0 ppt

(1σ) with 1 second time resolution.


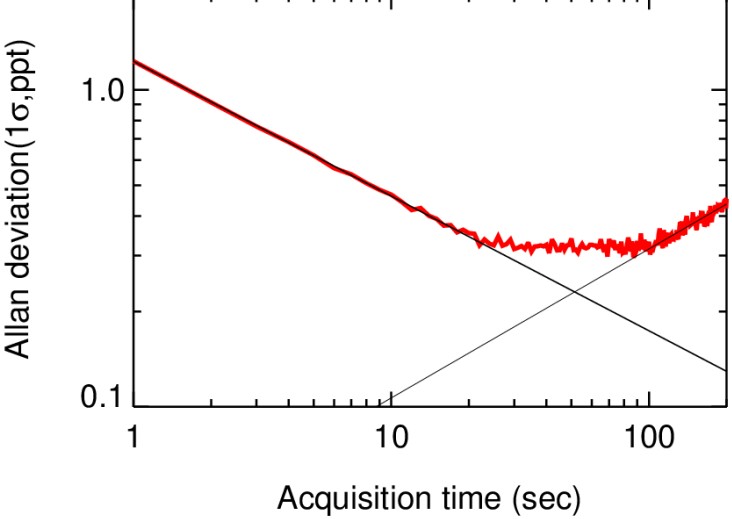

**Figure. 10** Allan deviation plots for measurements of $N_2O_5$. 12000 zero spectrums was measured in the laboratory with 1s integration time.

The best limit of the detection of this instrument is 2.1 ppt in the integral time of 50 s.




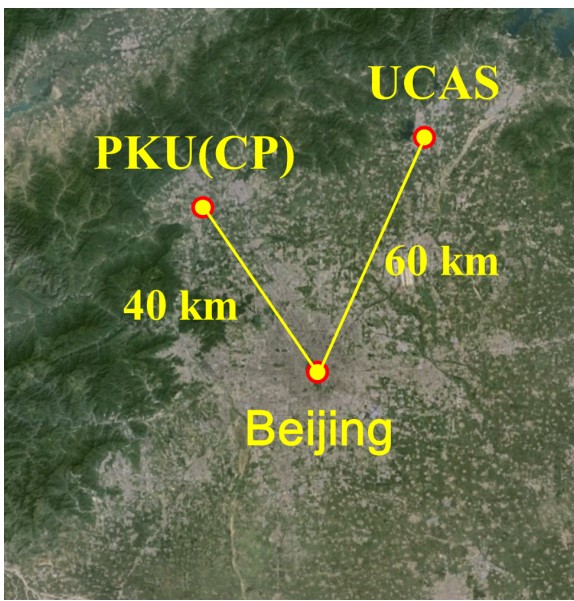

**Figure.11** Map of the UCAS winter campaign 2016, the UCAS site and PKU (CP) site is about 60 km and 40 km far from the center of

Beijing, respectively.

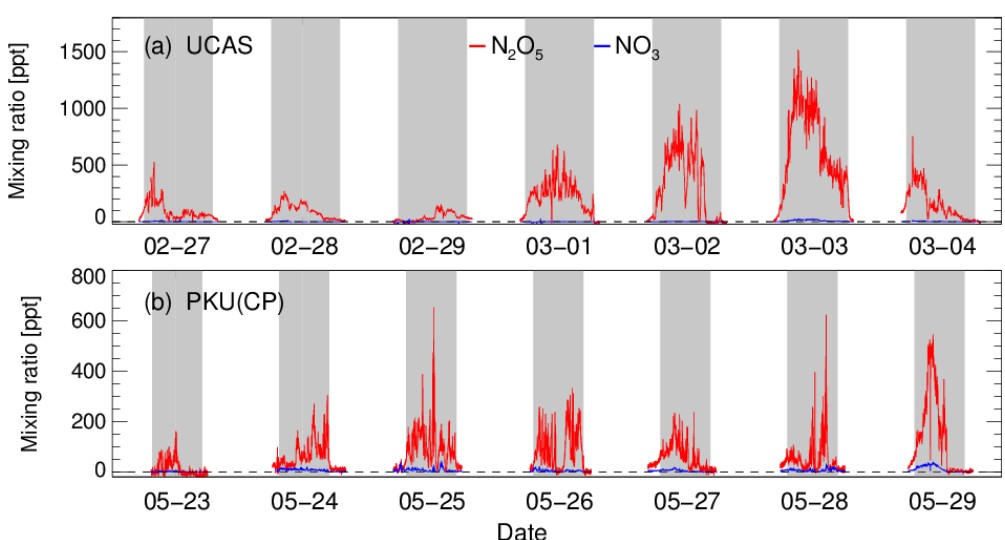

**Figure.12** An example of time series of $N_2O_5$ during the UCAS winter campaign 2016 and the PKU (CP) spring campaign, the red line

denoted $N_2O_5$ and the blue line denoted the calculated $NO_3$ based on steady state calculations, all the data was averaged to 1 minute. Panel (a)

shows a typical development of the observed $N_2O_5$ concentrations from clean to polluted air masses from 26 February to 4 March 2016 at

UCAS, panel (b) shows a typical pollution episode characterized during the PKU (CP) observations from 22 May to 29 May 2016.