# Peer review of "Development of a portable Cavity Enhanced Absorption Spectrometer for the measurement of ambient N2O5: experimental setup, lab characterizations, and field applications under polluted urban environment"

_Atmospheric Measurement Techniques, 2016_

## Referee Comment (RC1) · Anonymous Referee #1 · 7 Nov 2016

Review of Wang et al.: Development of a portable Cavity Enhanced Absorption Spectrometer for the measurement of ambient N2O5: experimental setup, lab characterizations, and field applications under polluted urban environment.

The manuscript of Wang et al., which reports on their new instrument to measure N2O5 is difficult to read. The level of English language is inadequate and only someone very close to this sort of instrument and its operation will make sense (after several readings) of some passages of text.

The manuscript offers little that can be considered more than repetition of that which is already found in the literature for similar instruments. There is no "significant" design innovation and indeed the use of just one cavity to measure the sum of NO3 and N2O5 means that this instrument can only be operated under high NOx (or low temperature) conditions when the NO3-to-N2O5 ratio is likely to be low.

The instrument is described as small and portable, yet no information about its weight or size (or power consumption) or given. No useful comparison is made to existing devices that measure N2O5.

The manuscript is not suitable for publication in AMT. The following comments may help the authors should they consider re-writing. They should also seek assistance in improving the English.

L25. Why do high levels of N2O5 imply an active nighttime chemistry ? If N2O5 builds up it may in part be due to lack of reactivity of NO3 or lack of uptake of N2O5 to aerosol.

L64. CEAS is suggested to give better selectivity that CRDS. Can the authors give an example of when CRDS measurements of NO3 are not specific. ?

L70. Vertical profiles of NO3 are suggested to be important. This is undoubtable the case, but why is it mentioned here ? is the instrument designed for or suitable for airborne operation (weight, power, size) ?

L90. Compare this instrument with the IBBCEAS already in operation (Langridge, Benton, Kennedy) ? Compare LOD and uncertainty with other N2O5 detection methods.

L148. Coated stand steel tube (stainless ?)

L187. "N2O5 is normally two orders of magnitude small concentrated than NO2 during nighttime. This is not true. There are plenty of examples where N2O5 is a substantial fraction of NOx. Also, N2O5 (and thus the NO3) formed can be close to zero at night (as the authors show in their own data). When N2O5 is close to zero, the NO3 formed by thermal dissociation is then not the dominant absorber.
L195. How was the deff established to be 45.0 cm using NO2  $?\,$  Was this a bottled standard of

NO2. What is the uncertainty of this approach (bottled mixing ratios, NO2 cross sections)?

L237. "The excess NO is sufficient to chemically destroy (destruct) the .....NO3." What was the NO mixing ratio, show the calculation. What was the NO2 impurity in the NO bottle ?

L295-313. The whole section is confusing. Some points: The purge flow does not result in a dilution of the NO3. It flushes the NO3 through the cavity changing the optical path length. This requires a different calculation to make the correction. Stopping the flow to measure NO3 loss in the cavity will mean that you lose information about point losses in front of the cavity (i.e. at mixing points in the tubing). Show the calculations to derive the effective transmission from the measurements of wall loss and residence time.

L311. How was the total transmission efficiency of NO3 derived ? What is the loss rate constant in cold PFA piping (the inlet) ?

L320. Explain how the best limit of detection was derived. Was it taken from the intersection of the two dotted lines ? Why should this give the best detection limit ?

L324. The total uncertainty on the scattering cross sections of He and N2 is given as 5 %. Where does this number come from ? Any significant difference between N2 and air ?

L344. What were the "conditions experienced for the winter campaign" that ensure that N2O5 / NO3 is greater than 10 ?

L355. The presence of HDV results in loss of N2O5. Provide (and justify) a hypothesis why this is the case.

AMTD
L364. What is meant by "a steady state calculation"? Is this referring thermodynamic equilibrium between NO2, NO3 and N2O5?

---

## Referee Comment (RC2) · Anonymous Referee #2 · 15 Nov 2016

This manuscript describes the construction, testing, and initial deployment of a cavityenhanced spectrometer for the detection of N2O5. The instrument is quite similar to prior NO3 / N2O5 instruments in the literature, although this instrument might be more portable or operate at reduced power. The manuscript should do a better job differentiating what was done by the authors from prior work. The mechanism of alignment may be novel, but that is not fully clear based upon the short description. The writing of the manuscript is a major problem, and the authors need to improve that aspect of

the manuscript to make it potentially acceptable for Atmospheric Measurement Techniques (AMT). The authors have clearly built a functional N2O5 instrument, but the manuscript needs improvement and better clarity of how this work is novel to be acceptable for AMT.

General issues:

Throughout the manuscript there are small missing details that should be included. Often these are things like the manufacturer / part number for components in the instrument (e.g. the filter, coating for cavity tube, "corrugated pipe", etc.) Please expand on these details so that one could have full details.

The discussion of d\_eff in section 3.2 is confusing. This seems to indicate that the effective length of the cavity differs between NO2 and NO3, which would be strange. Potentially there is some change in the purge between the configuration where NO2 and NO3 were measured?

It is preferred to use "mixing ratio" (rather than concentration) as the term for the ppt abundance of N2O5 (or any other chemical). Additionally, use of pmol mol-1 is preferred as more clear than ppt. For gases in the ppb range, one would use nmol mol-1.

The manuscript describes a good laboratory test for the inlet filter transmission, but does not describe how often the inlet filter is changed in operation, or if that change is based upon mass loading of the filter or simply a time criterion. Please explain operational filter change procedures. Some discussion of the decay of filter transmission, how that is quantified, and how the filter transmission decay affects the overall instrumental accuracy should be included.

The manuscript mentions the comparison to an Aerodyne I- CIMS, but indicates that comparison will come in a future publication. If this publication doesn't show any CIMS data, then it should not mention that CIMS data. Without any evidence shown of what "good agreement" is, this manuscript cannot make such a statement. I think that inclu-

AMTD
sion of the I- CIMS comparison would enhance this manuscript.

It is necessary to give some description of the instrument itself (e.g. size, weight, power consumption), and a photograph of the instrument would also be informative. Such a photograph would also probably answer question about the physical construction of the instrument.

It is not clear if NO3 is calculated from presumption of equilibrium and then corrected for or simply ignored. Please re-work discussion and presentation of NO3 in this manuscript.

Specific issues:

Line 38: Most areas with major NOx loadings also have significant aerosol loadings. Is this section meaning to indicate pollution aerosol, or potentially aerosol from nearby natural sources (e.g. desert dust)?

Line 46: Two ion CIMS ion chemistries have been demonstrated, NO3- and I-. Others could be possible, so the wording should be altered. Also, listing the reagent ion consistently is important.

Section 2.2 is listed twice on page 4 - it is both "optical layout" and "flow system".

Line 119: Please explain the "pilot experiments". Is there no flow cell in the middle? Are the mirrors on adjustable mounts, or how are they adjusted?

Line 126: What does "(-0.1) mm" mean?

Line 133: Maybe "homocentric" is "concentric"?

Line 148: "Stand" is maybe "Stainless"? How was the tube coated?

Line 173: Wording quite awkward here.

Line 234: Please clarify what is mean by "negative absorption of NO2". I understand this, but it could be made more clear to a general reader.
Line 296: I'm not clear on what is meant by ", which is limited by the transmission factor 2"? Clarify.

Line 343: This section discusses NO3, which apparently was not measured, but the figure shows a calculation of NO3. Please clarify that NO3 is only every calculated (e.g. in Fig. 12). It appears that the signals detected are simply interpreted as N2O5 without any correction for potential NO3, but that is made less clear by Eq.3, which seems to include NO3 in the observed signal. Please make this section more clear.

Fig. 6 caption: There are no labels as to what a), b) and c) mean. Is c) a residual or a non-detection of N2O5? The caption says "two spectra"

---

## Author Comment (AC1) · 27 Jan 2017

Response to Referees

Manuscript Number: amt-2016-319

Manuscript Title: Development of a portable Cavity Enhanced Absorption Spectrometer for the measurement of ambient $N_2O_5$: experimental setup, lab characterizations, and field applications under polluted urban environment

The discussion below includes the complete text from the referees, along with our responses to the specific comments and the corresponding changes made to the revised manuscript.

The detailed answers to the individual referee's comments in blue.

All of the line numbers refer to the original manuscript.

**Response to Referee #1 Comments:**

We would like to thank the referee for his/her detailed comments and suggestions which helped us a lot to improve the quality of the paper. Our revised manuscript has been further edited by professional language services.

1. The manuscript of Wang et al., which reports on their new instrument to measure $N_2O_5$ is difficult to read. The level of English language is inadequate and only someone very close to this sort of instrument and its operation will make sense (after several readings) of some passages of text. The manuscript offers little that can be considered more than repetition of that which is already found in the literature for similar instruments. There is no "significant" design innovation and indeed the use of just one cavity to measure the sum of $NO_3$ and $N_2O_5$ means that this instrument can only be operated under high NOx (or low temperature) conditions when the $NO_3$-to-$N_2O_5$ ratio is likely to be low.

   **Answer:**

   We agree with the referee that we need to better differentiate our instruments to previous ones. To our knowledge, there were only two other field deployable CEAS instruments which had been reported before. One is a LED based CEAS system from the Cambridge University group (Langridge et al., 2008; Benton et al., 2010; Kennedy et al., 2011) and the other is a laser based CEAS system which is an optional mode of a CRDS system from the Max Planck Institute of Chemistry group (Schuster et al., 2009; Crowley et al., 2010). Our system is also a LED based CEAS system. To make it field deployable, the instrument is featured with a novel non-adjustable mechanically aligned mirror mounts (the concentricity error < 0.01$^{o}$), see added Fig. 1b and Fig. 1c. The cavity is coupled automatically after we installed the HR mirrors into the mirror mounts and it reduces the background drift (e.g. thermal drift of the mechanics) significantly. The instrument is also featured with the addition of a chemical titration module which set up a dynamic reference spectrum for enhancing the precision of the spectrum fitting (e.g. the effects of the non-linear absorption lines of the water vapor is removed, the drift of the light power is eliminated). The chemical titration method has been widely used in the CRDS method for the determination of its zero point, but this is first time to be used in the CEAS type of instrument.

   We agree that the amount of $NO_3$ detected in our instrument is the sum of $NO_3$ and $N_2O_5$ in general and represents $N_2O_5$ under high NOx (or low temperature) conditions. We changed the

term $N_2O_5$ to be $NO_{3X}$ ($NO_{3X} = NO_3 + N_2O_5$) throughout the manuscript, the later one was referred to Stone et al., (2014).

[Figure]

**Figure 1.** A schematic plot of the newly developed IBBCEAS instrument for the detection of $NO_{3X}$. **(a)** overview of the optical layout (LEDs, collimating optics, high-finesse cavity, and spectrometer) and the flow system (aerosol filter, inlet, NO titration module, preheating tube, and detection cell). **(b)** the schematic layout of the mirror mounts, which enables a mechanical alignment of the high reflectivity (HR) mirrors. **(c)** a photograph of the mirror mounts. **(d)** the schematic layout of the NO titration module; the red arrow denotes the N2 gas flow, and the blue arrow denotes the NO gas flow.

2. The instrument is described as small and portable, yet no information about its weight or size (or power consumption) or given. No useful comparison is made to existing devices that measure $N_2O_5$. The manuscript is not suitable for publication in AMT. The following comments may help the authors should they consider re-writing. They should also seek assistance in improving the English.

**Answer:**

In the first graph of Sect. 2, we added a brief introduction as the followings:

"The total weight is less than 25 kg, approximate dimensions of 95×40×25 cm, the power consumption is less than 300 W."

We added a revised Table 3 in which a detailed comparison of the LOD and uncertainty with the existing field devices is presented (see also our answers to question 6.).

3. L25. Why do high levels of $N_2O_5$ imply an active nighttime chemistry ? If $N_2O_5$ builds up it may in part be due to lack of reactivity of $NO_3$ or lack of uptake of $N_2O_5$ to aerosol.
**Answer:**
The high levels of $NO_{3X}$ were observed in parallel with the presence of high aerosol loadings so that an active nighttime chemistry is implied.
We revised the sentence as: "Up to 1.0 ppb $NO_{3X}$ were observed with the presence of high aerosol loadings which indicates an active nighttime chemistry running in Beijing."

4. L64. CEAS is suggested to give better selectivity that CRDS. Can the authors give an example of when CRDS measurements of $NO_3$ are not specific. ?
**Answer:**
We agree with the referee. Both the CEAS and CRDS are suitable to detect $NO_3/N_2O_5$ with high selectivity. We removed this statement. In the past, we mean that CEAS measured the absorption spectrum with a wavelength window while CRDS observed cavity decay time caused by the molecules absorption at a specific wavelength.

5. L70. Vertical profiles of $NO_3$ are suggested to be important. This is undoubtable the case, but why is it mentioned here ? is the instrument designed for or suitable for airborne operation (weight, power, size) ?
**Answer:**
The instrument is designed with the feature of small size and low power consumptions which potentially meet the future applications on the mobile platforms for the vertical profile measurement. We revised the text accordingly.
We added the information about the weight, power and size in the first graph of Sect. 2 as suggested (see also our answers to question 2).

6. L90. Compare this instrument with the IBBCEAS already in operation (Langridge, Benton, Kennedy) ? Compare LOD and uncertainty with other $N_2O_5$ detection methods.
**Answer:**
We compared the LOD and uncertainty with other field $NO_3$ and $N_2O_5$ detection systems in Sect. 4.4 and Table 3 in the revised text as "The LOD and the uncertainty of the existing field measurement techniques of $NO_3$ and $N_2O_5$ ($NO_{3X}$) are listed in Table 3. For the $NO_3$ measurement, CRDS, CEAS and LIF are available with the LOD of 0.2 - 10 ppt and the uncertainties lower than 25%; for the $N_2O_5$ measurement, the three methods mentioned above and CIMS are available with the LOD of 0.5-12 ppt and the uncertainties lower than 40%. "

**Table 3.** Limits of detection (LOD) and uncertainty of the existing field deployable instruments of $NO_3$ and $N_2O_5$ ($NO_{3X}$).

| | Method | $NO_3$ | | $N_2O_5$ ($NO_{3X}$) | |
| --- | --- | --- | --- | --- | --- |
| | | LOD | Uncertainty | LOD | Uncertainty |
| This work | CEAS | | | 2.4 pptv (1s) | 19% - 22 % |
| Kennedy et al., 2011 | CEAS | 1.1 pptv (1s) | 11 % | 2.4 pptv (1s) | 14 % |
| Bitter et al., 2005 | CEAS | 1 pptv (100s) | | | |

| | | | | | |
|---|---|---|---|---|---|
| Schuster et al., 2009 | CRDS/ CEAS | 2 pptv (5s) | 14 % | 2 pptv (5s) | 13 % |
| Nakayama et al., 2008 | CRDS | 1.5 pptv (100s) | | | |
| Dube et al., 2006 | CRDS | 0.2 pptv (1s) | 25 % | 0.5pptv (1s) | 20 % - 40 % |
| Ayers et al., 2005 | CRDS | 2 pptv (25s) | | | |
| Wang et al., 2015 | CRDS | 3.2 pptv (10s) | 8 % | | |
| Matsumoto et al., 2006 | LIF | 10 pptv (600s) | 17 % | 12 pptv (600s) | 17 % |
| Slusher et al., 2004 | CIMS | | | 12 pptv (1s) | |
| Kercher et al., 2009 | CIMS | | | 2.7 pptv (60s) | 20 % |
| Wang et al., 2016 | CIMS | | | 4 pptv (60s) | 20 % |

7. L148. Coated stand steel tube (stainless?)

   **Answer:** We revised the "Coated stand steel tube" as "coated stainless tube"

8. L187. "$N_2O_5$ is normally two orders of magnitude small concentrated than $NO_2$ during nighttime. This is not true. There are plenty of examples where $N_2O_5$ is a substantial fraction of NOx. Also, $N_2O_5$ (and thus the $NO_3$) formed can be close to zero at night (as the authors show in their own data). When $N_2O_5$ is close to zero, the $NO_3$ formed by thermal dissociation is then not the dominant absorber.

   **Answer:**

   We agree with the referee and this assumption is actually not required in our data analysis. We deleted this sentence.

9. L195. How was the $d_{eff}$ established to be 45.0 cm using $NO_2$ ? Was this a bottled standard of $NO_2$. What is the uncertainty of this approach (bottled mixing ratios, $NO_2$ cross sections)?
   **Answer:**

   We supplied a $NO_2$ gas standard (200 ppb) with a constant flow into the cavity and then retrieved the $d_{eff}$ based on Eq. 1 in the text. The $NO_2$ gas standard was delivered from a bottled standard of $NO_2$ and diluted by synthetic air with a gas calibrator (TE-146i). The measured $NO_2$ concentration by switching off the purge flow was in good agreement with the delivered $NO_2$ gas standard (within 2%). The uncertainty of the $NO_2$ standard is estimated to be 2%, the uncertainty of the $NO_2$ cross section is estimated to be 4.7 % according to Voigt et al. (2002) and the associated uncertainty of the determined $d_{eff}$ with this approach is about 5%.

10. L237. "The excess NO is sufficient to chemically destroy (destruct) the …....$NO_3$." What was the NO mixing ratio, show the calculation. What was the $NO_2$ impurity in the NO bottle ?
    **Answer:**

    When NO injection is performed, the NO mixing ratio resulted in the sample gas flow is 480 ppb and the $NO_2$ impurity in the NO bottle is determined to be 0.8%. In the revised Sect. 2.3 "Dynamic reference spectrum", we addressed this issue as follows:

      "The NO titration module is connected to the inlet tube by a PFA tee-piece. Using a computer controlled solenoid valve, the instrument measures reference and sampling spectrums

sequentially by switching the NO injection on and off (NO = 98 ppmv, flow rate = 10 ml/min). A high purity $N_2$ line (OD = 3.175 mm) is added at the exit of the solenoid valve by a PFA tee-piece to flush the residual NO after the NO injection is switched off (Fig. 1d). The resulting NO mixing ratio is about 480 ppbv in the sample flow when NO injection is performed. Since 8.0 ppb $N_2O_5$ was once observed and reported in Hong Kong (Wang et al., 2016) as an extreme case, the ambient $NO_3$, $N_2O_5$, and $O_3$ were set at about 1 ppbv, 10 ppbv, and 100 ppbv, respectively, for the simulation, proving that the ambient $NO_3$ and $N_2O_5$ can be removed within a time scale of 0.05 s when NO is injected (Fig. 2).

The $NO_2$ impurity in the used NO standard is analysed by a commercial NOx instrument (TE-42i). The $NO_2$ impurity is found to be around 0.8 %, which means 4 ppbv of $NO_2$ is present in the reference spectrum measurement with the presence of 480 ppbv NO. The $NO_3$ and $O_3$ in the preheating tube and detection cell react with the high concentration of NO and generate $NO_2$. In the case shown as Fig. 2, the additional $NO_2$ produced during the measurement of the reference spectrum can reach up to 55 ppb (with the initial additional $NO_2$ set at 4 ppb). Therefore, to use this dynamic reference spectrum, we normally fit both $NO_3$ and $NO_2$ to cover the limiting cases when the generated $NO_2$ is high. Nevertheless, the fitted $NO_2$ concentration will be negative since the $NO_2$ concentrations are higher in the reference spectrum."

[Figure]

**Figure 2.** Simulation of the change of the mixing ratios of $NO_3$, $N_2O_5$ and $NO_2$ during the NO titration mode in the preheating tube and detection cell for an extremely high $NO_3$ and $N_2O_5$ case. The initial ambient $NO_3$, $N_2O_5$ and $O_3$ were set at 1 ppb, 10 ppb and 100 ppb, respectively. The initial $NO_2$ was set at 4 ppb from the impurity of the used NO standard.

11. L295-313. The whole section is confusing. Some points: The purge flow does not result in a dilution of the $NO_3$. It flushes the $NO_3$ through the cavity changing the optical path length. This requires a different calculation to make the correction. Stopping the flow to measure $NO_3$ loss in the cavity will mean that you lose information about point losses in front of the cavity (i.e. at mixing points in the tubing). Show the calculations to derive the effective transmission from the measurements of wall loss and residence time.

**Answer:**

The impact of the purge flow on the calculated $NO_3$ wall loss reactivity is now corrected through a view of changed optical path length.

The corresponding text is revised as follows:

"To determine the wall loss reactivity of $NO_3$, the heated detection cell is used as a flow tube. Gas samples with a stable amount of $N_2O_5$ are delivered by the $NO_3/N_2O_5$ source described above. By stopping the sample gas flow, the observed $NO_3$ versus the elapsed time determines the first order loss rate of $NO_3$ in the heated detection cell. In this experiment, the fitted first order uptake coefficient of $NO_3$ reflects the contribution from three processes: (1) the wall loss of the $NO_3$ in the detection cell; (2) the change of the effective cavity length due to the adding of the purge flows; and (3) the production of $NO_3$ from the reaction of $NO_2$ and $O_3$. The $NO_2$ concentration determined in the running sampling gas flow is used to determine the change of $d_{eff}$ corresponding to the elapsed time after stopping the sample flow (in the way it is used to quantify the $d_{eff}$ in Sect. 3.3). A time series of $d_{eff}$ is determined with high time resolution data acquisition (0.5 s) that is then used to quantify the mixing ratio of $NO_3$ in the corresponding time intervals. Figure 7 shows the decay of the observed $NO_3$ concentrations versus the elapsed time on a logarithmic scale. The fitted first order decay rate is $0.13 \pm 0.02$ s$^{-1}$ with a good correlation coefficient ($R^2$=0.991). Finally, the fitted first order decay rate is corrected by the chemistry of R1 and R4 with a box model constrained to observed $NO_2$ and $O_3$. The $NO_3$ wall reactivity of the heated detection cell surface is determined to $0.16 \pm 0.02$ s$^{-1}$, which is similar to previous results of 0.1–0.3 s$^{-1}$ (Brown et al., 2002; Crowley et al. 2010; Kennedy et al. 2011; Wang et al., 2015)."

The detailed calculations to derive the transmission efficiency of $NO_3$ and $N_2O_5$ due to the wall losses and the residence time are referred to our answers to question 12.

The surface materials are the same of the inlet tube, the preheating tube and the detection cell. Therefore, to determine the wall loss reactivity of $NO_3$ in the detection cell shall then be applicable for the inlet and the preheating tubes (As shown in Fig. 1, our instrument had only one mixing point at the set up of the NO titration module. And there is no block of the main sample gas flow of the PFA tee-piece. Thus, we think the influence of mixing point could be neglected).

12. L311. How was the total transmission efficiency of $NO_3$ derived ? What is the loss rate constant in cold PFA piping (the inlet) ?

   The detailed calculations to derive the total transmission efficiency of $NO_3$ and $N_2O_5$ are now listed as Table1 in the revised manuscript.

   As reported by Kennedy et al., (2011), the $NO_3$ wall loss reactivity in the cold PFA piping (inlet) is the same as the heated ones with the value of 0.27 s$^{-1}$. Nevertheless, we noticed that Crowley et al. (2010) reported that the $NO_3$ wall loss reactivity of the cold PFA tube could be a factor of two larger than that of the heated tube. We thus assumed our $NO_3$ wall loss reactivity for the cold PFA tube is between 0.16 s$^{-1}$ and 0.32 s$^{-1}$ and on average $NO_3$ wall loss reactivity for the cold PFA tube is estimated to be 0.24 s$^{-1}$ with an uncertainty of 0.06 s$^{-1}$.

**Table 1.** The transmission efficiency of $NO_3$ and $N_2O_5$ for the sampling module setup for the developed instrument.

| Gases | Filter | Inlet tube (0.7 s) | Preheating tube (0.14 s) | Cavity (0.46 s) | Total |
|---|---|---|---|---|---|
| $NO_3$ | $72\pm3$ %[a] | $84\pm4$ % ($k=0.24$ s$^{-1}$)[b] | 98 % ($k=0.16$ s$^{-1}$) | 93 % ($k=0.16$ s$^{-1}$) | $55\pm6$ % |
| $N_2O_5$ | $93\pm3$ %[a] | 99 % ($k=0.019$ s$^{-1}$) | $99\pm1$ %[c] | 93 % ($k=0.16$ s$^{-1}$) | $85\pm3$ % |

Note: [a] filter aging contributed an uncertainty of 3 %; [b] the uncertainty of the $NO_3$ wall loss reactivity in the cold inlet tube caused an uncertainty of 4 %; [c] the location of the $N_2O_5$ dissociation in the preheating tube had an uncertainty of 1 %.

13. L320. Explain how the best limit of detection was derived. Was it taken from the intersection of the two dotted lines? Why should this give the best detection limit ?
**Answer:**
We first used the Allan deviation plot to determine the best integration time and then analyzed the standard deviation of our measurement results for synthetic air at such integration time interval. According to the Allan variance approach, the best integration time appeared at the intersection of the white noise and the fitted drift (e.g. Fig. 6 of Langridge et al., 2008; Fig.8 of Min et al., 2016).

14. L324. The total uncertainty on the scattering cross sections of He and $N_2$ is given as 5 %. Where does this number come from ? Any significant difference between $N_2$ and air ?
**Answer:**
The total uncertainty on the scattering cross sections of $N_2$ is about 5% according to Sneep and Ubachs (2005) and the uncertainty for He makes a negligible contribution (Washenfelder et al., 2008). No significant difference is between $N_2$ and air, therefore air and He is also used to determine the mirror reflectivity during field studies (e.g. Min et al., 2016).

15. L344. What were the "conditions experienced for the winter campaign" that ensure that $N_2O_5$ / $NO_3$ is greater than 10 ?
**Answer:**
In the winter campaign, the averaged nighttime temperature and $NO_2$ mixing ratio were -4.3 °C and 15.5 ppb, respectively; the calculated ratio of $N_2O_5$ and $NO_3$ based on a box model was larger than 100.

16. L355. The presence of HDV results in loss of $N_2O_5$. Provide (and justify) a hypothesis why this is the case.
**Answer:**
It is known that HDV would emit large amount of fresh NO. The emitted NO will titrate both $O_3$ and $NO_3$ and reduce the accumulation of $N_2O_5$ ($NO_3+NO_2\rightarrow N_2O_5$) or enhance the loss of $N_2O_5$ ($N_2O_5\rightarrow NO_3+NO_2$) in the time scale of $N_2O_5$ thermal dissociation (0.1 – 20 min from summer to winter).

17. L364. What is meant by "a steady state calculation" ? Is this referring thermodynamic equilibrium between $NO_2$, $NO_3$ and $N_2O_5$ ?
**Answer:**
Yes, "a steady state calculation" refers to the thermodynamic equilibrium between $NO_2$, $NO_3$

and $N_2O_5$. We modified the text accordingly.

---

## Author Comment (AC2) · 27 Jan 2017

Response to Referees

Manuscript Number: amt-2016-319

Manuscript Title: Development of a portable Cavity Enhanced Absorption Spectrometer for the measurement of ambient $N_2O_5$: experimental setup, lab characterizations, and field applications under polluted urban environment

The discussion below includes the complete text from the referees, along with our responses to the specific comments and the corresponding changes made to the revised manuscript.

The detailed answers to the individual referee's comments in blue.

All of the line numbers refer to the original manuscript.

**Response to Referee #2 Comments:**

We would like to thank the referee for his/her detailed comments and suggestions which helped us a lot to improve the quality of the paper. Our revised manuscript has been further edited by professional language services.

1. This manuscript describes the construction, testing, and initial deployment of a cavity enhanced spectrometer for the detection of $N_2O_5$. The instrument is quite similar to prior $NO_3/N_2O_5$ instruments in the literature, although this instrument might be more portable or operate at reduced power. The manuscript should do a better job differentiating what was done by the authors from prior work. The mechanism of alignment may be novel, but that is not fully clear based upon the short description.

   The writing of the manuscript is a major problem, and the authors need to improve that aspect of the manuscript to make it potentially acceptable for Atmospheric Measurement Techniques (AMT). The authors have clearly built a functional $N_2O_5$ instrument, but the manuscript needs improvement and better clarity of how this work is novel to be acceptable for AMT.

   **Answer:**

   We agree with the referee that we should do a better job to differentiate what we have done compared to that from the prior work. To our knowledge, there were only two other field deployable CEAS instruments which had been reported before. One is a LED based CEAS system from the Cambridge University group (Langridge et al., 2008; Benton et al., 2010; Kennedy et al., 2011) and the other is a laser based CEAS system which is an optional mode of a CRDS system from the Max Planck Institute of Chemistry group (Schuster et al., 2009; Crowley et al., 2010). Our system is also a LED based CEAS system. To make it field deployable, the instrument is featured with a novel non-adjustable mechanically aligned mirror mounts (the concentricity error $< 0.01^\circ$), see added Fig. 1b and Fig. 1c. The cavity is coupled automatically after we installed the HR mirrors into the mirror mounts and it reduces the background drift (e.g. thermal drift of the mechanics) significantly. The instrument is also featured with the addition of a chemical titration module which set up a dynamic reference spectrum for enhancing the precision of the spectrum fitting (e.g. the effects of the non-linear absorption lines of the water vapor is removed, the drift of the light power is eliminated). The chemical titration method has been widely used in the CRDS method for the determination of its zero point, but this is first time to be used in the CEAS type of instrument.

[Figure]

**Figure 1.** A schematic plot of the newly developed IBBCEAS instrument for the detection of $NO_{3X}$. **(a)** overview of the optical layout (LEDs, collimating optics, high-finesse cavity, and spectrometer) and the flow system (aerosol filter, inlet, NO titration module, preheating tube, and detection cell). **(b)** the schematic layout of the mirror mounts, which enables a mechanical alignment of the high reflectivity (HR) mirrors. **(c)** a photograph of the mirror mounts. **(d)** the schematic layout of the NO titration module; the red arrow denotes the N2 gas flow, and the blue arrow denotes the NO gas flow.

2.  General issues: Throughout the manuscript there are small missing details that should be included. Often these are things like the manufacturer / part number for components in the instrument (e.g. the filter, coating for cavity tube, "corrugated pipe", etc.) Please expand on these details so that one could have full details.

**Answer:**

We agree with the comments and added the information of manufacturer / part number accordingly when needed throughout the manuscript. With respect to the filter, coating for cavity tube, and "corrugated pipe", we revised as follows in the revised manuscript:

Sect. 2.1: "The optical cavity is enclosed by a sample gas detection cell with a sample inlet, outlet, and two welded corrugated pipes connected at two ends."

Sect. 2.2: "A Teflon polytetrafluoroethylene (PTFE) filter (25 μm thickness, 4.6 cm diameter, 2.5 μm pore size, Typris, China) is used in the front of the sampling module to remove ambient aerosols…".

Sect. 2.2: "In Fig. 1b, the central part of the detection cell is constructed using a 35.6 cm long

PFA tube (marked in light green) (Entegris, I.D. =10 mm), enclosed by a stainless tube (marked in grey)."

3. The discussion of deff in section 3.2 is confusing. This seems to indicate that the effective length of the cavity differs between $NO_2$ and $NO_3$, which would be strange. Potentially there is some change in the purge between the configuration where $NO_2$ and $NO_3$ were measured?
   **Answer:**
   The effective length of the cavity ($d_{eff}$) describes the length occupied by the sample gas flow in the cavity which is the same for both $NO_2$ and $NO_3$. In our measurement, the $d_{eff}$ is determined to be 45 cm, 90% of the length of the cavity. Moreover, the distance between inlet and outlet ($d_{sample}$) is 39 cm (78% of the length of the cavity). The difference between the $d_{eff}$ and $d_{sample}$ showed that there was diffusion of sample gases into the purge volumes. Since the possibility of this diffusion is slow relative to the rate of the $NO_3$ wall losses, the determination of the $d_{eff}$ for $NO_3$ is associated with an additional uncertainty of 12%.

4. It is preferred to use "mixing ratio" (rather than concentration) as the term for the ppt abundance of $N_2O_5$ (or any other chemical). Additionally, use of pmol mol^-1 is preferred as more clear than ppt. For gases in the ppb range, one would use nmol mol^-1.
   **Answer:**
   We used "mixing ratio" in combination with the term of "pptv" as suggested.

5. The manuscript describes a good laboratory test for the inlet filter transmission, but does not describe how often the inlet filter is changed in operation, or if that change is based upon mass loading of the filter or simply a time criterion.
   Please explain operational filter change procedures. Some discussion of the decay of filter transmission, how that is quantified, and how the filter transmission decay affects the overall instrumental accuracy should be included.
   **Answer:**
   According to previous works, we changed the filter with a regular time interval (once per hour) to reduce the uncertainty from the decay of the filter transmission during pollution episodes. For clean conditions, the filter exchange frequency was reduced to be once per two hours when we observed much slower aerosol accumulation.
   During the campaign, we also stored the used filter and determined the $NO_3$ and $N_2O_5$ transmission efficiency afterwards in the lab. The transmission efficiencies of $NO_3$ and $N_2O_5$ are determined to be $72 \pm 3\%$ and $93 \pm 3\%$ due to the filter aging.
   In the revised manuscript, we added the description about the filter aging and the effect to the filter transmission as follows in the Sec.4.2.1.
   "The filter transmission efficiency of $NO_3$ and $N_2O_5$ is determined through the differentiation of an inlet without a filter, with a clean filter (25 μm thickness, 4.6 cm diameter, 2.5 μm pore size, Typris, China), and with used filters saved during typical pollution episodes during field measurements. According to previous field measurements of $NO_3$ and $N_2O_5$ (e.g. Brown et al., 2001; Schuster et al., 2009), frequent filter change is suggested, and the frequency is proposed to be 0.5–3 h depending on the aerosol loadings to reduce the impact of the filter aging caused by aerosol accumulation. For this reason, we changed the filter with a regular time interval

(once every hour) during pollution episodes. For clean conditions, the filter exchange frequency was reduced to be once every two hours.

For the determination of the $NO_3$ filter transmission efficiency, an additional preheating tube is inserted in front of the detection system to convert all the generated $NO_{3X}$ delivered by the calibration source to $NO_3$. The determined clean filter transmission efficiency is 75 % for $NO_3$ and is slightly lower than the previous results of $NO_3$ transmission efficiency on Teflon filters (Aldener et al., 2006; Schuster et al., 2009). The filter transmission efficiency of $NO_3$ on used filters is determined to be 5 % less than that on the clean filter. For the field calculation of the $NO_3$ concentrations, the filter transmission efficiency is then estimated to be 72 ± 3 %. For the determination of the $N_2O_5$ filter transmission efficiency, the mixing ratio of $NO_2$ and $O_3$ is modulated to achieve a high ratio of $N_2O_5/NO_3$ (>100) before being fed into the instrument. The transmission efficiency of the $N_2O_5$ on the clean filter is determined to be 96 %, which is consistent with the previous studies on the filter loss of $N_2O_5$ (Fuchs et al., 2008; Aldener et al., 2006; Schuster et al., 2009). The filter transmission efficiency of $N_2O_5$ on a used filter is determined to be 6 % smaller than that on the clean filter. Therefore, the filter transmission factor for $N_2O_5$ is estimated to be 93 ± 3 %."

6. The manuscript mentions the comparison to an Aerodyne I-CIMS, but indicates that comparison will come in a future publication. If this publication doesn't show any CIMS data, then it should not mention that CIMS data. Without any evidence shown of what "good agreement" is, this manuscript cannot make such a statement. I think that inclusion of the I- CIMS comparison would enhance this manuscript.

**Answer:**

During the campaign, an Aerodyne I-CIMS (Breton et al., 2012;Breton et al., 2014) from Gothenburg University Group was deployed in parallel with our instrument, but the CIMS did not perform in-situ $N_2O_5$ calibration during the campaign since it was originally targeted on the measurement of the gas phase precursors for SOA. The preliminary comparison showed reasonable agreements between the observed $NO_{3X}$ from CEAS and the raw intensities signal of $I(N_2O_5)^-$ from the Aerodyne I-CIMS. Since the results of the Aerodyne I-CIMS were not independently calibrated, we skipped the comparison toward the I-CIMS.

7. It is necessary to give some description of the instrument itself (e.g. size, weight, power consumption), and a photograph of the instrument would also be informative. Such a photograph would also probably answer question about the physical construction of the instrument.

**Answer:**

In the first graph of Sect. 2, we added a brief introduction as the followings:

"The total weight is less than 25 kg, approximate dimensions of 95×40×25 cm, the power consumption is less than 300 W."

Moreover, we added both a detailed schematic plot and a corresponding photograph about the details of our mirror mounting parts (cf. Fig. 1b and 1c presented in the answers to question 1).

8. It is not clear if $NO_3$ is calculated from presumption of equilibrium and then corrected for or simply ignored. Please re-work discussion and presentation of $NO_3$ in this manuscript.

**Answer:**

In the revised manuscript, we re-worked discussion and presentation of $NO_3$ as indicated by both referee #1 and #2.

We agree that the amount of $NO_3$ detected in our instrument during field studies is the sum of $NO_3$ and $N_2O_5$ in general and represents $N_2O_5$ under high NOx (or low temperature) conditions. We changed the term $N_2O_5$ to be $NO_{3X}$ ($NO_{3X}$ = $NO_3+N_2O_5$) throughout the manuscript, the later one was referred to Stone et al., (2014). And in this way, the calculation of $NO_3$ from presumption of equilibrium was not needed.

9. Specific issues: Line 38: Most areas with major NOx loadings also have significant aerosol loadings. Is this section meaning to indicate pollution aerosol, or potentially aerosol from nearby natural sources (e.g. desert dust)?

    **Answer:**

    We revised the sentence to be: "From satellite observations, it was found that the USA, Europe, and China are the three major high NOx regions worldwide (e.g. Richter et al., 2005). Moreover, in the North China Plain areas, the high NOx air masses often overlap with high aerosol loadings from both secondary formations as well as nearby natural sources (e.g. dust from the Gobi Desert in the spring) and serve as an ideal air mass for samples for the study of $NO_{3X}$ chemistry."

10. Line 46: Two ion CIMS ion chemistries have been demonstrated, NO3- and I-. Others could be possible, so the wording should be altered. Also, listing the reagent ion consistently is important.

    **Answer:**

    The introduction of CIMS on the Line 46 is revised as suggested:

    "In addition to optical approaches, different chemical ionization mass spectrometry (CIMS) methods have been used for the detection of ambient $N_2O_5$ (Slusher et al., 2004; Fortner et al. 2004; Kercher et al., 2009; Chang et al., 2011). Slusher et al. (2004) utilized ion reaction ($I^- + N_2O_5 \rightarrow NO_3^-$) to detect $N_2O_5$ at 62 amu ($NO_3^-$). Nevertheless, this approach showed cross sensitivity towards $NO_3$ ($I^- + NO_3 \rightarrow NO_3^-$) and additional interference from species like $ClONO_2$ and $BrONO_2$. A strong unknown interference at 62 amu was found for the detection of $N_2O_5$ under a high NOx regime in Hong Kong (Wang et al., 2014). Kercher et al. (2009) introduced an ion-molecule region (IMR) module wherein the ion reaction, $I^- + N_2O_5 \rightarrow I(N_2O_5)^-$, is enhanced so that $N_2O_5$ can be detected specifically at 235 amu. With this method, a direct measurement of $N_2O_5$ is achieved, showing a good comparison with the well-established CRDS system in Hong Kong (Wang et al., 2016)."

11. Section 2.2 is listed twice on page 4 – it is both "optical layout" and "flow system".

    **Answer:**

    Revised accordingly.

12. Line 119: Please explain the "pilot experiments". Is there no flow cell in the middle? Are the mirrors on adjustable mounts, or how are they adjusted?

    **Answer:**

In the pilot experiments, we explored the relations between the observed light output signals towards the cavity lengths (with adjustable mounts) and the diameters of the flow cell (mimicked by two irises). We found that in general a shorter cavity can achieve higher light output, but the improvement is not significant. Considering the instrument size, we choose the cavity length to be 50 cm. Moreover, we found that the light output is reduced dramatically when the flow cell diameter is smaller than 10 mm. To achieve a compromise of a high light output which ensures a good SNR on the detector and a short gas sample residence time which ensures the high transmission efficiency of $NO_3$ over the detection cell, the diameter of the detection flow cell is then chosen to be 10 mm at the expense of some losses of the output signal.

13. Line 126: What does "(-0.1) mm" mean?

**Answer:**

On the Line 125 and 126, the statement of "25.0 (-0.1) mm" means that the diameter of the HR mirror (25.0 mm) with a negative machined error of 0.1mm according to the manufacturer report (Layertec GmbH, Mellingen, Germany). We modified the text to be "25.0 (+0.0/-0.1) mm".

14. Line 133: Maybe "homocentric" is "concentric"?

**Answer:**

On the Line 133 we replaced "homocentric" with "concentric" accordingly.

15. Line 148: "Stand" is maybe "Stainless"? How was the tube coated?

**Answer:**

It is "Stainless". The central part of the detection cell is made by a 35.6 cm long PFA tube (Entegris, I.D.=10 mm) , and the PFA tube is enclosed by a stainless tube.

16. Line 173: Wording quite awkward here.

**Answer:**

We rewrite the sentence as follows in the revised manuscript:

"The effective absorption cross section of the abundant ambient absorbers, $NO_3$ and $NO_2$, in this wavelength window of 640 - 680 nm needs to be determined to retrieve the concentrations of $NO_3$. Since we used a dynamic reference spectrum which contains the same amount of water vapor as that of the measured sample spectrum (cf. Sect. 2.3), the calculation and fitting of the strong non linear absorptions lines of $H_2O$ in this wavelength window is avoided. "

17. Line 234: Please clarify what is mean by "negative absorption of $NO_2$". I understand this, but it could be made more clear to a general reader.

**Answer:**

We revised the text as the followings:

"The $NO_2$ impurity in the used NO standard is analysed by a commercial NOx instrument (TE-42i). The $NO_2$ impurity is found to be around 0.8 %, which means 4 ppbv of $NO_2$ is present in the reference spectrum measurement with the presence of 480 ppbv NO. The $NO_3$ and $O_3$ in the preheating tube and detection cell react with the high concentration of NO and generate

NO$_2$. In the case shown as Fig. 2, the additional NO$_2$ produced during the measurement of the reference spectrum can reach up to 55 ppb (with the initial additional NO$_2$ set at 4 ppb). Therefore, to use this dynamic reference spectrum, we normally fit both NO$_3$ and NO$_2$ to cover the limiting cases when the generated NO$_2$ is high. **Nevertheless, the fitted NO$_2$ concentration will be negative since the NO$_2$ concentrations are higher in the reference spectrum.**"

We added a new Figure 2 in the revised text as follows:

[Figure]

**Figure 2.** Simulation of the change of the mixing ratios of NO$_3$, N$_2$O$_5$ and NO$_2$ during the NO titration mode in the preheating tube and detection cell for an extremely high NO$_3$ and N$_2$O$_5$ case. The initial ambient NO$_3$, N$_2$O$_5$ and O$_3$ were set at 1 ppb, 10 ppb and 100 ppb, respectively. The initial NO$_2$ was set at 4 ppb from the impurity of the used NO standard.

18. Line 296: I'm not clear on what is meant by ", which is limited by the transmission factor2"? Clarify.

    **Answer:**

    In the revised manuscript, we calculated the total transmission efficiencies for NO$_3$ and N$_2$O$_5$, respectively, and listed the contributions from different part of the sampling and detection modules in Table 1.

**Table 1.** The transmission efficiency of NO$_3$ and N$_2$O$_5$ for the sampling module setup for the developed instrument.

| Gases | Filter | Inlet tube (0.7 s) | Preheating tube (0.14 s) | Cavity (0.46 s) | Total |
|---|---|---|---|---|---|
| NO$_3$ | 72±3 %[a] | 84±4 % (k=0.24 s$^{-1}$)[b] | 98 % (k=0.16 s$^{-1}$) | 93 % (k=0.16 s$^{-1}$) | 55±6 % |
| N$_2$O$_5$ | 93±3 %[a] | 99 % (k=0.019 s$^{-1}$) | 99±1 %[c] | 93 % (k=0.16 s$^{-1}$) | 85±3 % |

Note: [a] filter aging contributed an uncertainty of 3 %; [b] the uncertainty of the NO$_3$ wall loss reactivity in the cold inlet tube caused an uncertainty of 4 %; [c] the location of the N$_2$O$_5$ dissociation in the preheating tube had an uncertainty of 1 %.

19. Line 343: This section discusses NO$_3$, which apparently was not measured, but the figure shows a calculation of NO$_3$. Please clarify that NO$_3$ is only every calculated (e.g. in Fig. 12). It appears that the signals detected are simply interpreted as N$_2$O$_5$ without any correction for potential

$NO_3$, but that is made less clear by Eq.3, which seems to include $NO_3$ in the observed signal. Please make this section more clear.

**Answer:**

We agree and we have now interpreted our measurement signal as $NO_{3X}$ which is more accurate in principle. Thus the calculation of $NO_3$ is not required any further. Further explanation is referred to our answers to question 8.

20. Fig. 6 caption: There are no labels as to what a), b) and c) mean. Is c) a residual or a non-detection of $N_2O_5$? The caption says "two spectra

**Answer:**

The panel c) represents the residuals of the panel a) and panel b). Red line denotes the residual of panel a) and blue line the residual of panel b).